# MULTI-TASK LEARNING FOR SEMANTIC PARSING WITH CROSS-DOMAIN SKETCH

## ABSTRACT

Semantic parsing, which maps a natural language sentence into a machine-readable representation of its meaning, is highly constrained by the limited annotated training data. Inspired by the idea of coarse-to-fine, we propose a general-to-detailed neural network (GDNN) by incorporating middle coarse cross-domain sketch (CDS) among utterances and their logic forms. For utterances in different domains, the General Network will extract CDS using an encoder-decoder model in a multi-task learning setup. Then for some utterances in a specific domain, the Detailed Network will generate the detailed target parts using a sequence-to-sequence architecture with advanced attention to both utterance and generated CDS. Our experiments show that compared to direct multi-task learning, CDS has improved the performance in semantic parsing task which converts users' requests into meaning representation language (MRL). We also use experiments to illustrate that CDS works by adding some constraints to the target decoding process, which further proves the effectiveness and rationality of CDS.

## 1 INTRODUCTION

Recently many natural language processing (NLP) tasks based on the neural network have shown promising results and gained much attention because these studies are purely data-driven without linguistic prior knowledge. Semantic parsing task which maps a natural language sentence into a machine-readable representation (Fan et al. (2017)), as a particular translation task, can be treated as a sequence-to-sequence problem (Dong & Lapata (2016)). Lately, a compositional graph-based semantic meaning representation language (MRL) has been introduced (Kollar et al. (2018a)), which converts utterance into logic form (action-object-attribute), increasing the ability to represent complex requests. This work is based on MRL format for semantic parsing task.

Semantic parsing highly depends on the amount of annotated data and it is hard to annotate the data in logic forms such as Alexa MRL. Several researchers have focused on the area of multi-task learning and transfer learning (Hakkani-Tür et al. (2016), Fan et al. (2017), Kollar et al. (2018b)) with the observation that while these tasks differ in their domains and forms, the structure of language composition repeats across domains (Herzig & Berant (2017)). Compared to the model trained on a single domain only, a multi-task model that shares information across domains can improve both performance and generalization. However, there is still a lack of interpretations why the multi-task learning setting works (Ruder (2017)) and what the tasks have shared. Some NLP studies around language modeling (Le & Mikolov (2014), Vaswani et al. (2017), Devlin et al. (2018)) indicate that implicit commonalities of the sentences including syntax and morphology exist and can share among domains, but these commonalities have not been fully discussed and quantified.

To address this problem, in this work, compared to multi-task learning mentioned above which directly use neural networks to learn shared features in an implicit way, we try to define these cross-domain commonalities explicitly as cross-domain sketch (CDS). E.g., *Search weather in 10 days* in domain *Weather* and *Find schedule for films at night* in domain *ScreeningEvent* both have action *SearchAction* and Attribute *time*, so that they share a same MRL structure like *SearchAction(Type(time@?))*, where *Type* indicates domain and *?* indicates attribute value which is copying from the original utterance. We extract this domain general MRL structure as CDS. Inspired by the research of coarse-to-fine (Dong & Lapata (2018)), we construct a two-level encoder-decoder by using CDS as a middle coarse layer. We firstly use General Network to get the CDS for every

utterance in all domains. Then for a single specific domain, based on both utterance and extracted CDS, we decode the final target with advanced attention while CDS can be seen as adding some constraints to this process. The first utterance-CDS process can be regarded as a multi-task learning setup since it is suitable for all utterances across the domains. This work mainly introducing CDS using multi-task learning has some contributions listed below:

1) We make an assumption that there exist cross-domain commonalities including syntactic and phrasal similarity for utterances and extract these commonalities as cross-domain sketch (CDS) which for our knowledge is the first time. We then define CDS on two different levels (action-level and attribute-level) trying to seek the most appropriate definition of CDS.

2) We propose a general-to-detailed neural network by incorporating CDS as a middle coarse layer. CDS is not only a high-level extraction of commonalities across all the domains, but also a prior information for fine process helping the final decoding.

3) Since CDS is cross-domain, our first-level network General Network which encodes the utterance and decodes CDS can be seen as a multi-task learning setup, capturing the commonalities among utterances expressions from different domains which is exactly the goal of multi-task learning.

## 2 RELATED WORK

### 2.1 SPOKEN LANGUAGE UNDERSTANDING

Traditional spoken language understanding (SLU) factors language understanding into domain classification, intent prediction, and slot filling, which proves to be effective in some domains (Gupta et al. (2006)). Representations of SLU use pre-defined fixed and flat structures, which limit its expression skills like that it is hard to capture the similarity among utterances when the utterances are from different domains (Kollar et al. (2018b)). Due to SLU's limited representation skills, meaning representation language (MRL) has been introduced which is a compositional graph-based semantic representation, increasing the ability to represent more complex requests (Kollar et al. (2018a)). There are several different logic forms including lambda-calculus expression (Kwiatkowski et al. (2011)), SQL (Zhong et al. (2018)), Alexa MRL (Kollar et al. (2018a)). Compared to fixed and flat SLU representations, MRL (Kollar et al. (2018a)) based on a large-scale ontology, is much stronger in expression in several aspects like cross-domain and complex utterances.

### 2.2 SEQUENCE-TO-SEQUENCE FOR SEMANTIC PARSING

Mapping a natural language utterance into machine interpreted logic form (such as MRL) can be regarded as a special translation task, which is treated as a sequence-to-sequence problem (Sutskever et al. (2014)). Then Bahdanau et al. (2015) and Luong et al. (2015) advance the sequence-to-sequence network with attention mechanism learning the alignments between target and input words, making great progress in the performance. Malaviya et al. (2018) explore the attention mechanism with some improvements by replacing attention function with attention sparsity. Besides, to deal with the rare words, Gu et al. (2016) incorporate the copy mechanism into the encoder-decoder model by directly copying words from inputs. Lately, many researchers have been around improving sequence-to-sequence model itself, in interpreting the sentence syntax information. Eriguchi et al. (2016) encode the input sentence recursively in a bottom-up fashion. Wu et al. (2017) generate the target sequence and syntax tree through actions simultaneously. Another aspect which has caught much attention is constrained decoding. Krishnamurthy et al. (2017) and Post & Vilar (2018) add some constraints into decoding process, making it more controllable. Dong & Lapata (2016) use the recurrent network as encoder which proves effective in sequence representation, and respectively use the recurrent network as decoder and tree-decoder. Krishnamurthy et al. (2017) employ the grammar to constrain the decoding process. Dong & Lapata (2018), believe utterance understanding is from high-level to low-level and by employing sketch, improve the performance.

### 2.3 MULTI-TASK LEARNING

For semantic parsing task especially in MRL format, it is expensive and time-consuming to annotate the data, and it is challenging to train semantic parsing neural models. Multi-task learning aims to

use other related tasks to improve target task performance. Liu & Lane (2016b) deal with traditional SLU piper-line network by jointly detecting intent and doing slot filling. Sogaard & Goldberg (2016) share parameters among various tasks, according to the low-level and high-level difference. Hershcovich et al. (2018) divide the representation network into task-specific and general which is shared during multi-task learning. Fan et al. (2017) and Herzig & Berant (2017) directly share the encoder or decoder neural layers (model params) through different semantic parsing tasks. In Kollar et al. (2018b), multi-task learning also mainly acts sharing the params of the network.

# 3 APPROACH

## 3.1 DEFINITION OF CROSS-DOMAIN SKETCH

For human language expressions, especially task-oriented requests, there exist commonalities across sentences including linguistic syntax and phrase similarity. They can be seen with general sentence templates. Table 1 shows some examples.

| Utterance | Sentence Template | CDS |
|---|---|---|
| Find restaurants nearby | (find what where) | SearchAction(Type(place@?)) |
| What is the weather in NYC | (what where) | SearchAction(Type(place@?)) |
| Tell me what movies tonight | (tell me what when) | SearchAction(Type(time@?)) |
| Play last year's music | (play when's what) | PlayAction(Type(time@?)) |
| Book today's restaurant downtown | (book when's what where) | BookAction(Type(place@?,time@?)) |

Table 1: Utterances' Commonalities.

Since sentence templates are too many, we try to leverage these common regularities more abstractly. We extract these invariant commonalities which are implicit across domains, and call them as cross-domain sketch (CDS) in a canonical way.

We define CDS in meaning representation language (MRL) format (action-object-attribute) and on two levels (**action-level** and **attribute-level**). Action-level CDS means to acquire the same action for utterances from different domains while the attribute-level CDS means to extract more detailed information. See examples in Table 1. Instead of extracting CDS from utterance directly, we try converting from logic form into CDS reversely, because it is easier to deal with structural logic form than utterance in natural language form. We analyze the dataset Snips and use a rule-based method to obtain CDS. We strip logic forms from domain-specific components and preserve domain-independent parts including general actions and attributes.

We do some statistics on the dataset Snips (Goo et al. (2018)) used in this paper. We convert attributes [object_type, movie_type, restaurant_type] into {object_type}, [object_name, restaurant_name, movie_name] into {object_name}, [year, timeRange] into {time}, [location_name, current_location] into {object_location}, [country, city] into {place}. All those attributes account for 55% of all attributes which indicate the existence and feasibility of CDS.

## 3.2 MODEL

Figure 1 shows our overall network, which contains a two-level encoder-decoder. The **General Network** encodes utterance and decodes cross-domain sketch (CDS). Since this process is domain-general, it can be done to all domains, which is a multi-task setup. The **Detailed Network** firstly encodes the CDS and the utterance, then it decodes the target result based on both utterance and CDS. This process is domain-dependent, so that it is a fine-tuning process in a specific domain.

### 3.2.1 PROBLEM DEFINITION

For an input utterance $u = u_1, u_2, ...u_{|u|}$, and its middle result cross-domain sketch (CDS) $c = c_1, c_2, ...c_{|c|}$, and its final result logic form $y = y_1, y_2, ...y_{|y|}$, the conditional probability is:

$$p(y|u,c) = \prod_{t=1}^{|y|} p(y_{t|y_{<t}}, u, c) \qquad p(c|u) = \prod_{t=1}^{|c|} p(c_{t|c_{<t}}, u) \qquad (1)$$

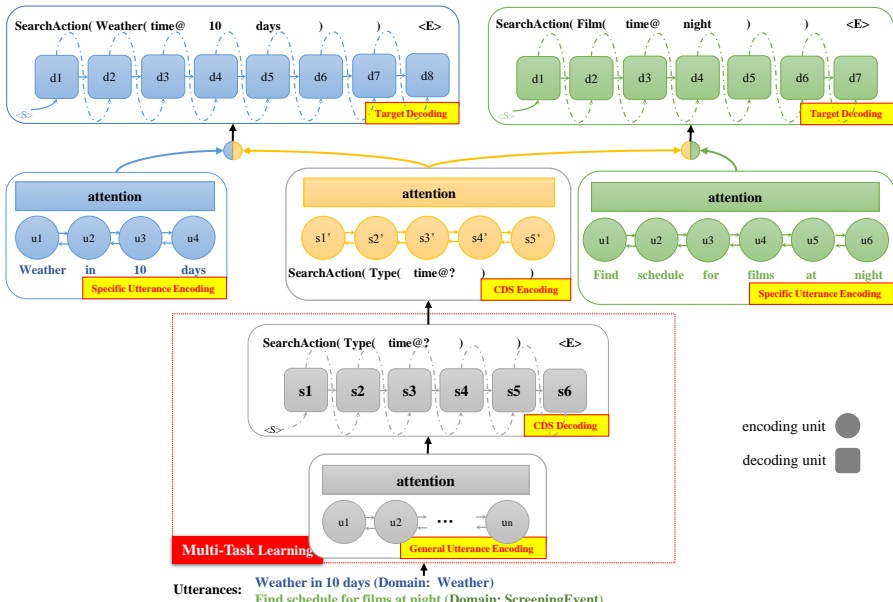

Figure 1: Overall Network. General Network (red dashed box below) encodes the utterance with bi-directional LSTM and decodes cross-domain sketch (CDS) using unidirectional LSTM with attention to utterance in all domains. For identical encoding, general utterance encoding and specific utterance encoding share the same encoder while for separate encoding, they are not (see Section 3.2.2). Then Detailed Network, in one specific domain, encodes CDS and utterance using bi-directional LSTM, decodes the final target with advanced attention to both utterance and CDS.

where $y_{<t} = y_1, y_2, ...y_{|t-1|}$, and $c_{<t} = c_1, c_2, ...c_{|t-1|}$.

### 3.2.2 UTTERANCE ENCODER

The neural encoder of our model is similar to neural machine translation (NMT) model, which uses a bi-directional recurrent neural network. Firstly each word of utterance is mapped into a vector $u_t \in \mathbb{R}^d$ via embedding layer and we get a word sequence $u = (u_1, ..., u_{|u|})$. Then we use a bi-directional recurrent neural network with long short-term memory units (LSTM) (Hochreiter & Schmidhuber (1997)) to learn the representation of word sequence. We generate forward hidden state $\overrightarrow{h_t^u} = f_{LSTM}(\overrightarrow{h_{t-1}^u}, u_t)$ and backward state $\overleftarrow{h_t^u} = f_{LSTM}(\overleftarrow{h_{t-1}^u}, u_t)$. The $t$-th word will be $h_t^u = [\overrightarrow{h_t^u}, \overleftarrow{h_t^u}]$.

We construct two kinds of utterance encoders, **general utterance encoder** for General Network and **specific utterance encoder** for Detailed Network (see in Figure 1), so as to extract different information for different purposes. The general utterance encoder, meant to pay more attention to cross-domain commonalities of utterances, is used by all utterances from all domains. The specific utterance encoder, which is domain-dependent, belongs to one specific domain and is more sensitive to details. We call encoder outputs $h_t^{ug}$ from general utterance encoder and $h_t^{us}$ from specific utterance encoder. When the two encoders share the same parameters that is $h_t^{ug} = h_t^{us}$, we call it **identical encoding** and when they are not, we call it **separate encoding**, inspired by (Sogaard & Goldberg (2016);Krishnamurthy et al. (2017);Liu et al. (2017);Abdou et al. (2018)) which explore the sharing mechanisms of multi-task learning and propose some improvements.

### 3.2.3 CDS DECODER & ENCODER

The General Network is meant to obtain cross-domain sketch (CDS) $c$ conditioned on utterance $u$, using an encoder-decoder network. After encoding utterance by general utterance encoder for all domains, we obtain $h_t^{ug}$(see Section 3.2.2). Then we start to decode CDS.

The decoder is based on a unidirectional recurrent neural network, and the output vector is used to predict the word. The $cd$ represents CDS decoder.

$$h_t^{cd} = f_{LSTM}(h_{t-1}^{cd}, c_t) \tag{2}$$

where $c_t$ is the previously predicted word embedding.

The LuongAttention (Luong et al. (2015)) between decoding hidden state $d_t$ and encoding sequence $e_i(i = 1, 2, ...|e|)$ at time step $t$ is computed as:

$$s_{t,i} = \frac{\exp(d_t, e_i)}{\sum_{k=1}^{|c|} \exp(d_t, e_k)} \tag{3}$$

$$a_t = \sum_{i=1}^{|e|} s_{t,i} e_i \tag{4}$$

Based on equations (3) and (4), we compute the attention $a_t^u$ by setting $d_t = h_t^{cd}$ and $e_i = h_i^{ug}(i = 1, 2, ...|u|)$. The $t$-th predicted output token will be:

$$h_{t_{att}}^{cd} = tanh(W_{cd} h_t^{cd} + W_{cu} a_t^u) \tag{5}$$

$$p(c_{t|c<t}, u) = softmax(W_{co} h_{t_{att}}^{cd} + b_{co}) \tag{6}$$

where $W, b$ are parameters. After decoding CDS words $c = (c_1, ..., c_{|c|})$, we use an encoder to represent its meaning and due to words' relation with forward and backward contexts, we choose to use a bi-directional LSTM. We generate forward hidden state $\overrightarrow{h_t^{ce}}$ and backward state $\overleftarrow{h_t^{ce}}$. The $t$-th word will be $h_t^{ce} = [\overrightarrow{h_t^{ce}}, \overleftarrow{h_t^{ce}}]$.

### 3.2.4 TARGET DECODER

Through specific utterance encoder and cross-domain sketch (CDS) encoder, we acquired $t$-th word representation $h_t^{us}$ and $h_t^{ce}$. Finally with advanced attention to both encoded utterance $u$ and CDS $c$, we decode the final target $y$. The decoder is based on a unidirectional recurrent neural network, and the output vector is used to predict the word. The $y$ represents target decoder.

$$h_t^y = f_{LSTM}(h_{t-1}^y, y_t) \tag{7}$$

where $y_t$ is the previously predicted word embedding. During target decoding process and at time step $t$, we not only compute the attention to utterance encoding outputs $h^{us}$ but also compute the attention to CDS encoding outputs $h^{ce}$. The attention between target hidden state and utterance is $a_t^u$ by computing attention between $h_t^y$ and encoding sequence $h_i^{us}(i = 1, 2, ...|u|)$ based on equations (3) and (4). The attention between target hidden state and CDS is $a_t^c$ by also computing attention between $h_t^y$ and $h_i^{ce}(i = 1, 2, ...|c|)$ in the same way. Then the $t$-th predicted output token will be based on the advanced two-aspect attention:

$$h_{t_{att}}^y = tanh(W_y h_t^y + W_{yu} a_t^u + W_{yc} a_t^c) \tag{8}$$

$$p(d_{t|d<t}, u) = softmax(W_{yo} h_{t_{att}}^y + b_{yo}) \tag{9}$$

### 3.3 MODEL TRAINING AND INFERENCE

For training process, the objective is:

$$max \sum_{(u,c,y) \in T} p(y|u, c) + p(c|u) \tag{10}$$

$T$ is the training corpus. For inference process, we firstly obtain cross-domain sketch (CDS) via $\widetilde{c} = argmax\, p(c|u)$ then we get the final target logic form via $\widetilde{y} = argmax\, p(y|u, \widetilde{c})$. For both decoding processes, we use greedy search to generate words one by one.

# 4 EXPERIMENTS

## 4.1 DATASETS

Existed semantic parsing datasets, e.g., GEO (Zettlemoyer & Collins (2012)), ATIS (Zettlemoyer & Collins (2007)), collect data only from one domain and have a very limited amount, which can not fully interpret the effectiveness of cross-domain sketch (CDS) since it needs large dataset among different domains.

In this case, we mainly consider the semantic parsing task Snips (Goo et al. (2018)) based on MRL format (action-object-attribute). Snips collects users' requests from a personal voice assistant. The original dataset is annotated in spoken language understanding (SLU) format (intent-slot). It has 7 intent types and 72 slot labels, and more statistics are shown in Table 2. Based on the format (intent-slot), we pre-process this dataset into MRL format by some pre-defined rules, then we regard the intent as domain/task and share CDS among them. The details are shown in Table 3.

| Domain | Total | AddTo Playlist | Book Restaurant | Get Weather | Play Music | Rate Book | Search CreativeWork | Search ScreeningEvent |
|--------|-------|---------|---------|---------|-------|------|-------------|----------------|
| train | 13084 | 1818 | 1881 | 1896 | 1914 | 1876 | 1847 | 1852 |
| dev | 700 | 100 | 100 | 100 | 100 | 100 | 100 | 100 |
| test | 700 | 124 | 92 | 104 | 86 | 80 | 107 | 107 |

Table 2: Statistics of the dataset Snips.

| Utterance | let me know the weather forcast of stanislaus national forest far in nine months |
|-----------|------|
| Intent | GetWeather |
| Slots | O O O O O O O B-geographic_poi I-geographic_poi I-geographic_poi B-spatial_relation O B-timeRange I-timeRange |
| Action-level CDS | SearchAction |
| Attribute-level CDS | SearchAction ( Type ( poi @ ? , spatial_relation @ ? , time @ ? ) ) |
| Target | SearchAction ( WeatherType ( geographic_poi @ 7 8 9 , spatial_relation @ 10 , timeRange @ 12 13 ) ) |
| Utterance | find the schedule for films at night at great escape theatres |
| Intent | SearchScreeningEvent |
| Slots | O O B-object_type O B-movie_type O B-timeRange O B-location_name I-location_name I-location_name |
| Action-level CDS | SearchAction |
| Attribute-level CDS | SearchAction ( Type ( object_type @ ? , movie_type @ ? , time @ ? , object_location @ ? ) ) |
| Target | SearchAction ( ScreeningEventType ( object_type @ 2 , movie_type @ 4 , timeRange @ 6 , location_name @ 8 9 10 ) ) |

Table 3: Several examples of Snips. Utterance is the user's request which is a natural language expression. Intent and slots are in formats from original dataset. Cross-domain sketch (CDS) has two levels (action-level and attribute-level). Target is the final logic form with numbers indicating copying words from utterance (index starting from 0).

## 4.2 SETTINGS

We use Tensorflow in all our experiments, with LuongAttention (Luong et al. (2015)) and copy mechanism. The embedding dimension is set to 100 and initialized with GloVe embeddings (Pennington et al. (2014)). The encoder and decoder both use one-layer LSTM with hidden size 50. We apply the dropout selected in $\{0.3, 0.5\}$. Learning rate is initialized with 0.001 and is decaying during training. Early stopping is applied. The mini-batch size is set to 16. We use the logic form accuracy as the evaluation metric.

## 4.3 RESULTS AND ANALYSIS

Firstly, in order to prove the role of the cross-domain sketch (CDS) in helping to guide decoding process with multi-tasking learning setup, we do several experiments, and the results are shown in Table 4. For joint learning, we apply several multi-task architectures from (Fan et al. (2017)), including one-to-one, one-to-many and one-to-shareMany. One-to-one architecture applies a single sequence-to-sequence model across all the tasks. One-to-many only shares the encoder across all the tasks while the decoder including the attention parameters is not shared. In one-to-shareMany model, tasks share encoder and decoder (including attention) params, but the output layer of decoder is task-independent.

From the Table 4, in general, joint learning performs better than single task learning. In joint learning, one-to-one is the best and performs way better than one-to-many and one-to-shareMany, probably limited by the dataset's size and similarity among tasks. By incorporating CDS, our GDNN (general-to-detailed neural network) models have all improved the performance to different degrees. The CDS is defined on two levels (action-level and attribute-level, see examples in Table 3) and attribute-level CDS improves greater than action-level CDS, which is in our expectation since it offers more information for tasks to share. We also experiment on different utterance encoding setups with identical encoding and separate encoding (see Section 3.2.2). The separate encoding setup performs better than sharing the same encoder for utterance, which integrates the fact that different encoders pay different attention to the utterances due to different purposes which means one is more general and the other is more specific detailed.

| Method | Snips Accuracy |
|---|---|
| Single Seq2Seq (Sutskever et al. (2014)) | 62.3 |
| Joint Seq2Seq (one-to-many) (Fan et al. (2017)) | 62.0 |
| Joint Seq2Seq (one-to-shareMany) (Fan et al. (2017)) | 64.2 |
| Joint Seq2Seq (one-to-one) (Fan et al. (2017)) | 71.4 |
| GDNN with Action-level CDS (identical encoding) | 74.9 |
| GDNN with Action-level CDS (separate encoding) | 75.1 |
| GDNN with Attribute-level CDS (identical encoding) | 76.7 |
| GDNN with Attribute-level CDS (separate encoding) | 78.1 |

Table 4: Multi-task Results. Single Seq2Seq means each task has a sequenece-to-sequence model. Joint Seq2Seq show results with three multi-task mechanisms. Our results include GDNN (general-to-detailed neural network) models with different levels of CDS (action-level/attribute level) and different utterance encoding mechanisms (identical encoding/separate encoding).

We also list the full results of GDNN in Table 5 below, including CDS accuracy by General Network, target accuracy by Detailed Network when feeding with right CDS, and the final accuracy, which further proves the effectiveness of CDS.

| GDNN | CDS Accuracy | Target Accuracy while CDS is true | Final Accuracy |
|---|---|---|---|
| Action-level CDS (identical encoding) | 100.0 | 74.9 | 74.9 |
| Action-level CDS (separate encoding) | 100.0 | 75.1 | 75.1 |
| Attribute-level CDS (identical encoding) | 93.7 | 81.8 | 76.7 |
| Attribute-level CDS (separate encoding) | 91.0 | 83.6 | 78.1 |

Table 5: GDNN Results. Full results of general-to-detailed neural network (GDNN) with different levels of CDS (action-level/attribute level) and different utterance encoding mechanisms (identical encoding/separate encoding).

Moreover, we compare our experiments with traditional models which regard the task as intent classification and slot filling (IC_SF). The results are shown in Table 6 below.

From Table 6, we can see compared to IC_SF models (based on sequence labeling format), Seq2Seq perform worse (71.4% compared to 73.2%) due to its fewer assumptions and larger decode size as well as its difficulty of training, which is usual in comparing IC_SF models and sequence-to-sequence models. Through using CDS, the performance has significantly improved. On the one

| Method | Snips Accuracy |
|---|---|
| Joint Seq. (Hakkani-Tür et al. (2016)) | 73.2 |
| Atten.-Based (Liu & Lane (2016a)) | 74.1 |
| Slot.-Gated (Intent Atten.) (Goo et al. (2018)) | 74.6 |
| Slot.-Gated (Full Atten.) (Goo et al. (2018)) | 75.5 |
| Joint Seq2Seq (Fan et al. (2017)) | 71.4 |
| GDNN with Action-level CDS | 73.2 |
| GDNN with Attribute-level CDS | 74.6 |

Table 6: Seq2Seq results vs traditional results. The first four results show IC_SF models' performance. The last three results are based on the Seq2Seq architecture.

hand, CDS extract the cross-domain commonalities among tasks helping to make the multi-task learning more specific, which can be seen as an advance to multi-task learning.

On the other hand, CDS can be seen adding some constraints to the final target decoding process which has offered more information for the decoding process, compared to direct joint Seq2Seq. To better prove and explain this idea, we do some experiments according to constraint decoding aspect. We try to compare the sub-process of converting utterance to CDS through different models, e.g., IC_SF, Seq2Seq. From the Table 7, we can see that Seq2Seq achieve the comparable results (87.7%) to IC_SF model (84.9%) for generating CDS from utterance, which further explains that, the fact joint seq2seq performs worse (71.4%, see Table 6) than IC_SF model (73.2%) is owing to the lack of guidance and constraints during the follow-up decoding process. By incorporating CDS, we add some constraints to this decoding process thus obtaining better performance.

| Method | | Attribute-level Accuracy |
|---|---|---|
| | intent | 97.3 |
| IC_SF | slot | 87.3 |
| | final | 84.9 |
| Seq2Seq | | 87.7 |

Table 7: Results of CDS generation in dataset Snips by two methods. IC_SF is using intent classification and slot filling with evaluation metric (intent accuracy, slot labelling accuracy and final accuracy). Seq2Seq generates CDS based on utterance using an encoder-decoder.

## 5 CONCLUSIONS AND FUTURE WORK

In this paper, we propose the concept of cross-domain sketch (CDS) which extracts some shared information across domains, trying to fully utilize the cross-domain commonalities such as syntactic and phrasal similarity in human expressions. We try to define CDS on two levels and give some examples to illustrate our idea. We also present a general-to-detailed neural network (GDNN) for converting an utterance into a logic form based on meaning representation language (MRL) form. The general network, which is meant to extract cross-domain commonalities, uses an encoder-decoder model to obtain CDS in a multi-task setup. Then the detailed network generates the final domain-specific target by exploiting utterance and CDS simultaneously via attention mechanism. Our experiments demonstrate the effectiveness of CDS and multi-task learning.

CDS is able to generalize over a wide range of tasks since it is an extraction to language expressions. Therefore, in the future, we would like to perfect the CDS definition and extend its' ontology to other domains and tasks. Besides, in this paper, we use attention mechanism to make use of CDS which is still a indirect way. We would like to explore more effective ways such as constraint decoding to further enhance the role of CDS.

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
