# OpenReview forum: "Multi-Task Learning for Semantic Parsing with Cross-Domain Sketch"
_ICLR.cc/2019/Conference_

### Official Review · AnonReviewer3 · 2018-11-02
**Semantic parsing model with a domain-agnostic intermediate decoding step**

**Rating:** 5
**Confidence:** 4

**Review:**

This paper describes a two-stage encoder-decoder model for semantic parsing. The model first decodes a cross-domain schema (CDS) representation from the input utterance, then decodes the final logial form from both the utterance and CDS. The model outperforms other multitask Seq2Seq models on the Snips (Goo et al., 2018) dataset, but is still behind the traditional slot-filling models (Goo et al., 2018).

My main concern is that it is unclear to me how CDS (cross-domain schema) can be generalized to the other semantic parsing datasets, e.g., the Overnight dataset (Wang et al., 2015), which also contains multiple domains.

I think it would be nice to have some details about the CDS in the paper. For example, I’m wondering 1) how is this CDS designed? 2) how are the CDS annotations derived from the target output?

There are other details missing regarding the comparisons and the evaluation metrics. In 4.2, the authors mentioned “We use accuracy as the evaluation metric’’, does “accuracy” mean full logical form accuracy or accuracy on execution results?

* More minor comments:
In the first paragraph of Section 3, “irrelevant to domain’’ -> “domain-general’’ or “domain-agnostic’’?

It will be nice to write something more specific than “explore more ways to make it work better” in the future work.

This paper has some grammatical errors and formatting issues (e.g. missing space before punctuations).

* Missing references:
Neural semantic parsing over multiple knowledge-bases, Herzig and Berant, ACL 2017 <- This paper explores shared encoder/decoder for multi-domain semantic parsing, which is very related.

(Concurrent) Decoupling Structure and Lexicon for Zero-Shot Semantic Parsing, Herzig and Berant, EMNLP 2018

---

> ### Author Response · Authors · 2018-11-25
> **Response to Reviewer3**
>
> Thank you for your reviews!
>
> 1, CDS definition and value
> In the revisioned paper, we add a subsection 3.1 to describe CDS definition in detail. We also list the attributes that can form CDS since actions are very few due to the limitation of the dataset. Besides, we do some statistics on the dataset and reveal the percent that CDS can share.
> CDS is an extraction of cross-domain features. CDS based ontology can be generalized to other semantic parsing datasets, as long as there are common expressions in the utterances, which can be manifested from the target side such as some logic form. In the dataset used in our work, we convert the target logic form into CDS by some pre-defined rules. As for other datasets, we can do the same process. The main concern is that we should know the dataset very well and find the common shared things (in our case, action/attribute) which put high demands on datasets' amount and quality.
>
> 2, evaluation accuracy
> We use the logic form accuracy as the evaluation metric. Since the logic form is in a tree structure, we compare the predict tree and ground truth tree which is the accuracy.
>
> 3, some clarifications
> The “irrelevant to domain’’ means  “domain-general’’, since we focus on the features across the domains. In the future work, we would like to explore more ways to make CDS work such as constraint decoding or other more direct ways since now we use the attention mechanism to incorporate CDS into the network.

---

### Official Review · AnonReviewer1 · 2018-11-03
**Where's SRL?**

**Rating:** 4
**Confidence:** 4

**Review:**

This is a wonderful paper as it seems to have brougth Semantic Role Labelling (SRL) in the context of DL and in the context of voice search.
Results are interesting but the paper has some major limitations. In fact, the paper totally disregard the work on Semantic Role Labelling and on languages for expressing the general meaning of language in terms of relations and in terms of concepts.

The first limitation is on the key idea. The key idea of the paper seems to be the existence of an intermediate representation language to encode meaning for utterances. Yet, this intermediate language seems to be the final language with the same relation types (for example, SearchAction) and without representations for the involved concepts (Type that becomes alternatively Film or Weather according to the target final language). This seems to be SRL where the first step is to recognize the relation and, then, the second step is to recognize the roles even if roles are slot filler types in the case of this papers.

The second limitation is on how the intermediate language has been choosen. What is the relation with FrameNet or VerbNet? Why the authors have not choosen something similar? What are the limitations of these two resources that have forced the authors to disregard them?

Minor problems
====
- Why there are not spaces between characters and opening brackets?
- "compositional graph based .... language" is a really large noun compound

---

> ### Author Response · Authors · 2018-11-25
> **Response to Reviewer1**
>
> Thanks for bringing the idea of SRL and other reviews.
>
> 1, CDS definition
> In the revisioned paper, we add a subsection 3.1 to describe CDS definition in detail. We also list the attributes that can form CDS since actions are very few due to the limitation of the dataset. Besides, we do some statistics on the dataset and reveal the percent that CDS can share.
> CDS is an extraction of cross-domain features. Compared to FrameNet or VerbNet, CDS is more abstract and more like a schema composition since it is an extraction of common info of language expressions and it is across the domains. In the dataset used in our work, we convert the target logic form into CDS by some pre-defined rules. As for other datasets, we can do the same process.
>
> 2, why not SRL
> Though SRL and semantic parsing seem very similar, SRL focuses more on syntax analysis while we try to get the intent and slots. For semantic parsing task using the dataset Snips, the sentences, which are users' requests collected from voice assistants, are usually imperative sentences without subjects, which are not suitable for SRL task. SRL is meant to label the predicates‘ arguments with given predicates, however, for this work, we should convert the whole sentence including predicates/objects into normalized pre-defined actions/attributes.
>
> 3, grammar and format
> Moreover, we did some refinements in grammar and formatting.

---

### Official Review · AnonReviewer2 · 2018-11-05
**Overall score 3**

**Rating:** 3
**Confidence:** 4

**Review:**

Overall Score 3

This paper introduces “Cross domain schemas” (CDS) for semantic parsing of utterances made to a virtual assistant. CDS captures similarities in requests according to the underlying actions or attributes being discussed, regardless of the user’s high-level intent. Also introduced is a model which leverages CDS to improve semantic parsing of utterances to a meaning representation language (MRL). This model first parses an utterance to CDS, then uses an encoding of the CDS jointly with the utterance encoding to decode a meaning representation. By treating different intents as separate domains, the authors construct a multi-task learning setup for CDS and MRL parsing. Results are provided for the Snips dataset of virtual assistant queries.

Unfortunately, this paper fails to sufficiently explain its main proposal, the CDS. The stated goal is to explicitly define the cross-domain features that would otherwise be implicitly learned by the parameters of a neural network, yet no explicit definition is given. No rough quantification of how many or what percent of features appear across domains is provided. Rather, significant time and space is given to describing a fairly unsophisticated two-decoder model for inserting the mysterious CDS representation into the final decoding task.

The paper ignores standard semantic parsing datasets (GeoQuery and ATIS) due to their size and scope. However, comparable models (Goo et al. 2018) are trained and tested on ATIS. Moreover, an evaluation on a small, unseen target domain would be the perfect justification for the kind of cross-domain learning proposed here.

Instead, this paper opts only to evaluate on the recent Snips dataset. This dataset seems to be best suited to evaluating intent classification and slot filling (intent-slot), but the current work fails to improve over what Goo et al. 2018 report on this data. In the current work, the Snips dataset is used to evaluate MRL parsing, where the CDS model shows improvements over other seq2seq models. However, since MRL can be parsed from intent-slot format by predefined rules, it is uncertain whether the CDS model outperforms the Goo et al. model at even the task of MRL parsing (no such comparison is provided).

Overall, the paper suffers from some clarity issues especially regarding the definition and value of CDS. The model provided may be slightly original but is quite similar to the model of Dong and Lapata 2018. The significance of this work is questionable due to the poor comparison with recently released baseline models for the more common intent-slot task.

Pros

Introduces “Cross Domain Schemas” (CDS) for semantic parsing, which help improve robustness of semantic parsers by allowing models to learn patterns in one domain for use in another.

Through the use of CDS, train semantic parsers in a multi-task learning setup

Cons

CDS is not described in sufficient detail. In particular, the possible actions and attributes are not defined.

The model is described as “multi-task learning”, however all tasks are parsing requests made of a virtual assistant.

Results on standard data for semantic parsing such as GEO or ATIS are not reported.

The model does not appear to improve the results on the Snips dataset compared to the paper that introduces this dataset. Thus, the value of CDS is difficult to judge.

No per-domain analysis of the impact of CDS is provided.

---

> ### Author Response · Authors · 2018-11-25
> **Response to Reviewer2**
>
> Thanks for the thoughtful review! We respond as follows.
>
> 1, CDS definition
> In the revisioned paper, we add a subsection 3.1 to describe CDS definition in detail. We also list the attributes that can form CDS since actions are very few due to the limitation of the dataset. We do some statistics on the dataset and reveal the percent that CDS can share.
>
> 2, why not choose ATIS or GEO
> In subsection 4.1 experiments, we explain why we do not choose ATIS or GEO because they collect data only from one domain and have a limited amount which is not suitable for cross-domain experiments.
>
> 3, multi-task learning
> For dataset Snips, each domain can be seen as a task, so we regard this experiment as multi-task learning. Since we suffer a lot from existed datasets, in the future, we would like to construct more suitable datasets, proposing the idea of CDS.
>
> 4, experiments
> We do MRL parsing task on the same conditions and compare our model(74.6) with Seq2Seq(one-to-one, 71.4). Compared to direct multi-task learning, our model improves the performance.

---

### Author Response · Authors · 2018-11-25
**Summary of revisions**

Thank you all for the valuable reviews. Based on your suggestions, we have made some updates to our paper and have uploaded it on openreview.net.

1, CDS definition
In the revisioned paper, we add a subsection 3.1 to describe CDS definition in detail. We also list the attributes that can form CDS since actions are very few due to the limitation of the dataset. Moreover, we do some statistics on the dataset and reveal the percent that CDS can share.

2, CDS value
CDS is an extraction of cross-domain features. Since CDS is defined from language ontology itself, it can be generalized to other semantic parsing tasks.

3, paper structure
We compressed the space for the two-layer encoder-decoder network and paid more attention to CDS. Our intuition is to propose the idea of CDS, so we focus on how it is defined and how it works by experimental analysis.

Furthermore, we add responses to different reviewers separately as follows.

---

### Meta-Review · Area_Chair1 · 2018-11-06
**Two stage approach for semantic parsing leveraging cross domain schemas**

**Confidence:** 4
**Recommendation:** Reject

**Metareview:**

Interesting approach aiming to leverage cross domain schemas and generic semantic parsing (based on meaning representation language, MRL) for language understanding. Experiments have been performed on the recently released SNIPS corpus and comparisons have been made with multiple recent multi-task learning approaches. Unfortunately, the proposed approach falls short in comparison to the slot gated attention work by Goo et al.

The motivation and description of the cross domain schemas can be improved in the paper, and for replication of experiments it would be useful to include how the annotations are extended for this purpose.

Experimental results could be extended to the other available corpora mentioned in the paper (ATIS and GEO).